# Evaluation of the impact of chemical control on the ecology of *Rattus norvegicus* of an urban community in Salvador, Brazil

Arsinoê Cristina Pertile[1,2], Ricardo Lustosa[3], Ticiana Carvalho-Pereira[2,3], Gabriel Ghizzi Pedra[2,4], Jesus Alonso Panti-May[2], Udimila Oliveira[2], Caio Graco Zeppelini[1], Fábio Neves Souza[1,2,3], Daiana S. Oliveira[2], Hussein Khalil[5], Mitermayer G. Reis[2,6,7], James Childs[6], Albert I. Ko[2,6], Mike Begon[4], Federico Costa[2,3,5,6]*

1 Programa de pós-graduação em Ecologia: Teoria, Aplicações e Valores, Instituto de Biologia, Universidade Federal da Bahia, Salvador, Brazil, 2 Instituto de Pesquisas Gonçalo Moniz, Fundação Oswaldo Cruz, Ministério da Saúde, Salvador, Brazil, 3 Instituto de Saúde Coletiva, Universidade Federal da Bahia, Salvador, Brazil, 4 Institute of Integrative Biology, University of Liverpool, Liverpool, United Kingdom, 5 Swedish University of Agriculture Sciences, Umea, Sweden, 6 Department of Epidemiology of Microbial Disease, Yale School of Public Health, New Haven, Connecticut, United States of America, 7 Faculdade de Medicina da Bahia, Universidade Federal da Bahia, Salvador, Brazil

* fcosta2001@gmail.com

**Data Availability Statement:** Data are available in Zenodo at https://doi.org/10.5281/zenodo.6672759.

## Abstract

### Background

The presence of synanthropic rodents, such as *Rattus norvegicus*, in urban environments generates high costs of prophylaxis and control, in large part due to the environmental transmission of the pathogenic spirochete *Leptospira interrogans*, which causes leptospirosis. In Salvador, Brazil, The Center for Control of Zoonosis (CCZ) is responsible for planning and implementing Rodent Control Programs (RCP) which are based on chemical rodenticide. However, these strategies have not been standardized for use in developing countries.

### Aim

This study aimed to identify the effect of a chemical control campaign on the demographic variables of urban *R. norvegicus*, analyzing relative abundance, sex structure, body mass, and age of the population, as well as the characterization of spatial distribution among households, rodent capture campaigns and interventions.

### Methods

This study was carried out during 2015 in three valleys of an urban poor community in Salvador. Individuals of *R. norvegicus* were systematically captured before (Pre-intervention) and three months (1st post-intervention) and six months (2nd post-intervention) after a chemical control intervention conducted by the CCZ in two valleys of the study area while the third valley was not included in the intervention campaign and was used as a non-intervention reference. We used analysis of variance to determine if intervention affected demographic

**Funding:** This work was supported by the Oswaldo Cruz Foundation and the Health Surveillance Secretariat of the Ministry of Health (grants R01 AI052473, U01 AI088752, R01TW009504, R25 TW009338 and R01 AI121207) and the Wellcome Trust (102330/ Z/13/Z and 218987/Z/19/Z). CGZ and FNS hold doctorate scholarships granted by the Foundation for Research Support of the state of Bahia (FAPESB). The funders had no role in study design, data collection and analysis, decision to publish, or preparation of the manuscript.

**Competing interests:** no competing interests

variables and chi-square to compare proportions of infested households (Rodent infestation index–PII).

## Results

During the chemical intervention, 939 households were visited. In the pre-intervention campaign, an effort of 310 trap nights resulted in 43 rodents captured, and in the 1st and 2nd, post-intervention campaigns resulted in 47 rodents captured over 312 trap nights and 36 rodents captured over 324 traps-nights, respectively. The rodent infestation index (PII) points did not show a reduction between the period before the intervention and the two periods after the chemical intervention (70%, 72%, and 65%, respectively). Regarding relative abundances, there was no difference between valleys and period before and two periods after chemical intervention (trap success valley 1: 0,18; 0,19; 0,18 / Valley 3 0,15; 0,17; 0,13/ P>0,05). Other demographic results showed that there was no difference in demographic characteristics of the rodent population before and after the intervention, as well as there being no influence of the application of rodenticide on the areas of concentration of capture of rodents between the campaigns.

## Conclusion

Our study indicates that the chemical control was not effective in controlling the population of *R. norvegicus* and provides evidence of the need for re-evaluation of rodent control practices in urban poor community settings.

## Introduction

Unplanned urbanization and the increase in human population have favored the establishment of poor urban settlements or slums, which are characterized by highly populated residential areas and a lack of essential public services, such as garbage collection, sewage systems, water supply, electricity, and public lighting [1]. According to UN-Habitat, currently, 1.6 billion people live under such conditions, with urbanization, leading to environmental degradation and growing inequality [2], with drastic impacts on hydrologic cycles, green area suppression, pollution, and climate change both local and global [3]. These changes often cause loss of native biodiversity and increased abundance of introduced and generalist species, such as synanthropic rodents [4].

Synanthropic rats are among the most impactful pests to human activity, causing economic impacts in the hundreds of billions of dollars worldwide [5], and can transmit zoonotic pathogens responsible for significant human morbidity and mortality around the world [6]. Among these pathogens, *Leptospira interrogans* is a bacterium that causes leptospirosis, an emerging infectious disease in urban centers in developing countries [7]. Each year more than one million cases occur worldwide causing approximately 60,000 deaths [7]. In Brazil, more than 10,000 severe cases of leptospirosis are reported during epidemic events in periods of high precipitation in urban poor communities [8]. Mortality for severe forms of the disease, such as Weil's syndrome is >10% [9], exceeding 50% for pulmonary hemorrhage syndrome [10]. In urban poor communities, the transmission of this disease is more intense due to the proximity between humans and rodent populations. In addition, the characteristics of urban poor

communities (e.g., open sewers, accumulated trash, dirt floors) create a habitat ideal for rats, which leads to high infestation rates and frequent contact with residents [11].

One of the main strategies of the Brazilian Ministry of Health, and many other developing countries, to prevent or reduce human exposure to leptospires is to control rodent reservoirs through chemical control, the most widely used method to eliminate rats on a large scale [12,13]. Studies on rats and possible eradication methods are largely concentrated in the USA and Europe, in northern temperate climates [14]. In these regions, control with rodenticides has proved effective in reducing the density of these animals by 50–90% [15]. Additionally, studies have shown that the demographic structure of rodent populations is altered by the removal of dominant individuals and the potential resulting immigration influx [6].

Few studies have evaluated the impact of chemical interventions on *Rattus norvegicus* populations and they have reported a high reduction in infestation rates from 69% to 100%. One exception was a study performed in London [16], where poisoning interventions reduced rat population sizes by 10% and those population sizes returned to prebaiting population estimates within six months. In a study in Brazil, in Sao Paulo, reductions in infestation rates of 64% and a general reinfestation rate of 80% six months after the intervention were reported [17]. However, most of these studies provide no data on the effects of rodent control on population characteristics such as the number of embryos, percent of pregnant females etc, and the few studies that report demographic data on rat populations are limited to temperate regions [14]. Considering that patterns of pathogen infection are shaped by population features of reservoir hosts [18–20], it is critical to understand the effect of chemical control on the abundance and demographic characteristics of *R. norvegicus*, the main reservoirs host of *L. interrogans* in Brazil [21]. This information can help us to establish effective control interventions, in order to reduce the risk of pathogen transmission to humans [22].

The objective of this study, therefore, was to evaluate the effects of a rodent chemical control campaign carried out by the Center for Control of Zoonosis (CCZ) in the city of Salvador on the relative abundance and demographic markers of *Rattus norvegicus* in an urban community with a high risk of leptospirosis transmission.

## Material and methods

### Ethics statement

The ethics committee for the use of animals from the Oswaldo Cruz Foundation, Salvador, Brazil approved the protocols used in this study (protocol number 003/2012) These protocols were also approved by the Yale University's Institutional Animal Care and Use Committee (IACUC), New Haven, Connecticut (protocol number 2012–11498).

### Study area

The study was conducted in the neighborhood of Pau-da-Lima, in the city of Salvador (BA, Brazil), which is 0.17 km$^2$ in extent (Fig 1A) with 3,717 residents [23]. The site is a slum area described in detail previously [23] and is characterized by three geographic valleys (referred to as valleys 1, 2, and 3) where there is no regular refuse collection and sanitary structure, with open sewers. The mean annual incidence of leptospirosis is 35.4 per 1000 inhabitants [24] and, with a high abundance of rodents (*Rattus norvegicus* is a more dominant species) [23] and a prevalence of *L. interrogans* in rats [25].

In 2015, the CCZ carried out rodent control activities in 11 areas of the city of Salvador with a high risk of leptospirosis, which was defined by the density of cases registered in the areas by the Municipal Secretariat of Health, as well as previous data on rodent infestation collected by the CCZ. These activities included the application of rodenticide, educational

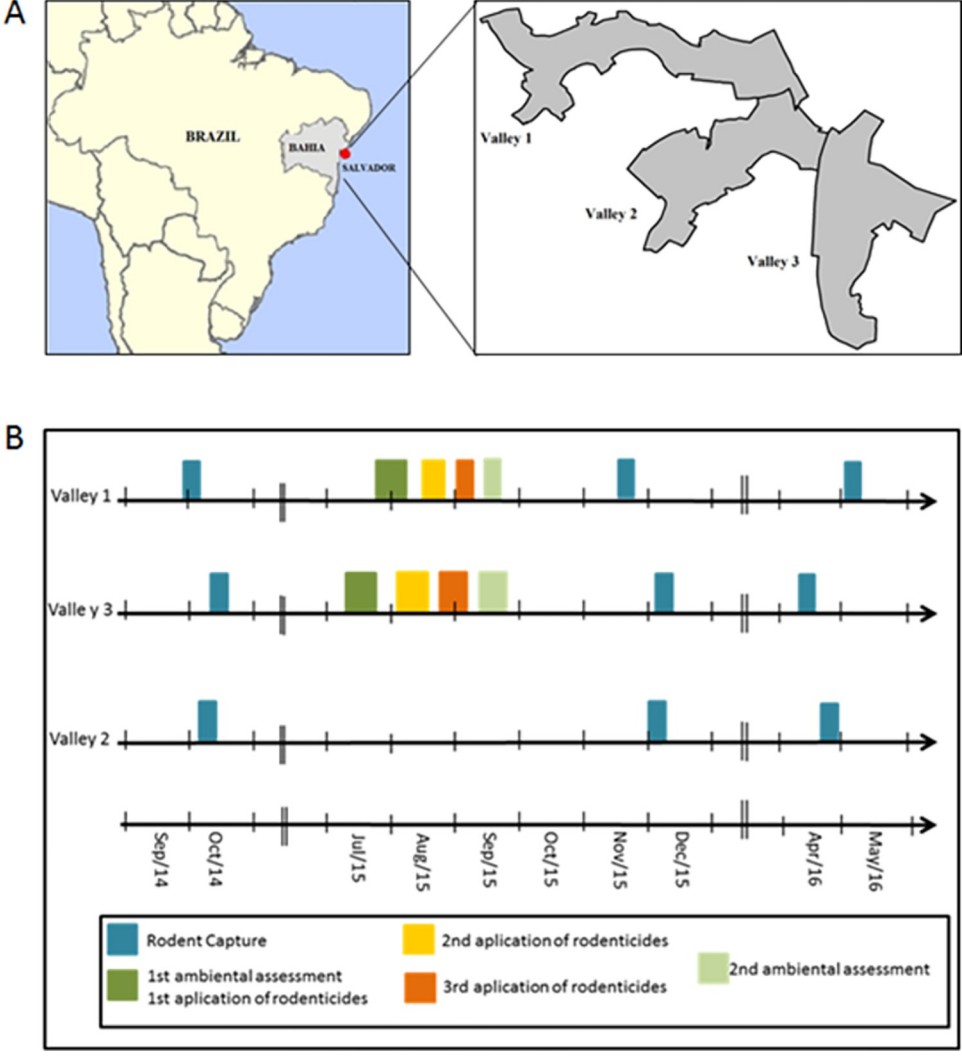

**Fig 1. A.** Study area in Brazil. **B**. Schedule of events in Pau da Lima, Salvador, Brazil. Capture, environmental assessment, chemical intervention stratified by valley during the study period.

campaigns, and environmental management. In the urban poor community of Pau da Lima, control activities focused mostly on chemical control which was performed in two of the three valleys. To evaluate this intervention, we performed rodent trappings before and after the intervention in valleys 1, 2, and 3 and monitored the CCZ chemical intervention performed in valleys 1 and 3. Valley 2 was used as a control area with no chemical intervention. (see Fig 2 for detailed steps).

### Rodent control intervention carried out by the Zoonoses Control Center

The CCZ carried out its intervention in the period from July 14 to September 15, 2015. As per its standard practice, the CCZ's intervention was performed in three steps: 1) an initial evaluation of the houses for rodent infestation (pre-intervention); 2) three rounds of rodenticide application (chemical intervention), and 3) evaluation of post-intervention infestation. This intervention follows standard methodologies described previously [26] and is used extensively by other CCZs throughout Brazil [12].

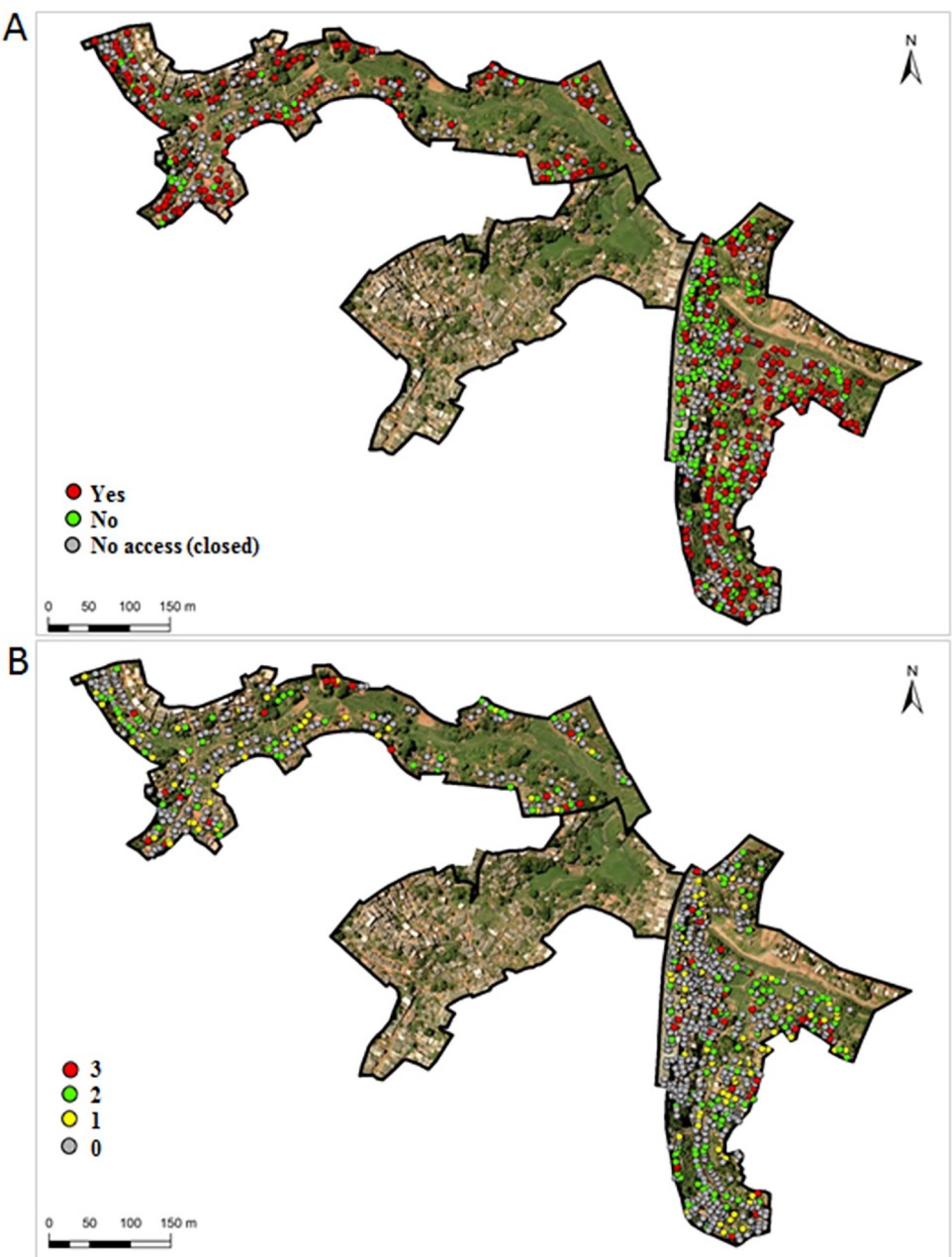

**Fig 2. A.** Area of study and distribution of households visited with or without need for rodenticide application in Pau da Lima, Salvador, Brazil. **B**. Number of rodenticide applications among households in need in Pau da Lima, Salvador, Brazil.

1. The pre-intervention assessment of rodent infestation was performed from July 14 to August 4, 2015. The CCZ team visited all abandoned houses, occupied housing, and public spaces in the study area. The team visited each household and classified them as 'inspected' when owners accepted the intervention, "abandoned", "refused" when owners rejected the intervention, or "closed" when owners were absent. In the inspected houses, a standardized questionnaire for the evaluation of rodent infestation adapted from CDC protocols [27] was applied as previously tested [28]. During the evaluation, 12 variables were assessed,

including the type of domicile, availability of food, water, access/entry for rodents to houses, and signs of rodent infestation (feces, burrows, trails, and grease marks on external walls).

2. Application of rodenticide (round 1). Simultaneously with the previous activity (evaluation of pre-intervention infestation), CCZ Endemy Control Agents (ECAs) applied rodenticides to households that showed signs of rodent infestation. The following commercial formulations were used: Coumatek® contact powder (Cumatetralil 0.75%) and Klerat paraffin block (Brodifacoum 0.005%). The contact powder was applied to the tracks and entrance of burrows. The baits were tied with pieces of wire so they would not be dragged by rodents or other animals. Rodenticides were only applied when the risk to children or domestic animals was excluded. The amount of rodenticide applied in each household depended on the area of the property, environmental conditions, and level of rodent infestation.

3. Application of rodenticide in rounds 2 and 3. Subsequent visits were performed at 10 (round 2) and 20 (round 3) days after round 1. The status of the baits (totally consumed, partially consumed, or not consumed) was recorded and new baits were placed according to the need.

4. Post-intervention rodent infestation evaluation. Fifteen days after round 3, the last visit was conducted to each household's, and the same questionnaire used for the initial evaluation was applied to evaluate the level of infestation after the intervention.

All these activities were carried out between 08:30 and 12:00 in the morning, in valleys 1 and 3. In valley 2 no visit and/or chemical control by the CCZ was carried out because the institution prioritized other risky areas for the prevention of leptospirosis. Therefore, in this study, this valley was used as a reference. Four people from the research team followed the CCZ teams during all stages–recording household classification, infestation, and rodenticide data–to ensure the quality of the information. The researchers had no influence on the nature or timing of the intervention.

### Rodent trapping and processing

We performed rodent trapping campaigns 10 months before CCZ's chemical intervention and three and six months after to evaluate demographic changes in the rodent populations (Fig 2).

The data collected during a previous study [23] conducted between October and December 2014 was used as pre-intervention trapping campaign. The previous study included trapping in 108 randomly selected sites in the three valleys of the study area. For the present study, we initially considered the points of the previous study in which trapping success was different from zero in order to guarantee the success of the analyses of abundance. Of these, 40 sampling points were randomly selected in the three valleys. Subsequently, we returned to the same 40 points to carry out new rodent trappings three- and six months post-intervention (November-December 2015 and March-April 2016, respectively).

During trapping campaigns, protocols already described were used [23,29]. The research team applied another questionnaire [28] on the 40 points to identify signs of rodent infestation (e.g.: feces, rodent runs, grease marks). In parallel, two Tomahawk traps were installed in each household within a 10-meter radius buffer. Trapped rodents were transported to the laboratory where they were euthanized. For each rat, demographic characteristics were recorded (sex and age structure, pregnancy, number of embryos, body mass/body condition) [19,23]. Rodent's body condition was estimated using a "scaled mass index" (SMI) [30]. All data were collected following previously validated methods using REDCap software [31].

## Data analyses

To calculate the infestation rate of valleys 1 and 3 we used the infestation index (IIP), a reference method utilised by the Brazilian Ministry of Health to estimate the level of vector infestation in urban centers [13] that uses number of houses with signs of rodent infestation/number of the households evaluated. To verify if there was variation between IIP before and after the intervention, a binomial probability test was used. To estimate the relative abundance of rats we uthe sed trap success rate. This index was calculated by dividing the total number of trapped rats by the total trap effort [32].

Differences in the means of age (days), SMI, and a number of embryos (response variables) of the trapped individuals between campaigns and valleys (independent variables) were first assessed through analysis of variance (ANOVA), followed by Tukey Honestly Significant Difference. Permutation ANOVA and Fisher's Least Significant Difference were the alternative tests when ANOVA test assumptions were not met. In addition, to determine if the sexual ratio varied from 1:1 between campaigns, a binomial probability test was used. We compared the proportion of pregnant females between the campaigns using a chi-square test of homogeneity.

To analyze the effect of the number of rodenticide baits on trapping success, a beta-inflated regression was used, which has a wide range of distribution shapes (Attachment 1) [33]. We also tested whether trap success, as a response variable, was associated with the proportion of household visits receiving rodenticide applications using a generalized linear model (glm) with a binomial distribution.

We assessed whether the need for rodenticide application (as a binomial response variable) was associated with the household distance to the main avenue (here used as a proxy for quality of infrastructure), accounting for the effect of the valley by applying a glm using household distance to the main avenue (continuous) and valley (factor) as explanatory variables.

Model simplifications were performed considering the Akaike Information Criterion (AIC). In all statistical analyses a significance level of $p < 0.05$ was considered and was performed in R [34] using packages lmPerm, agricolae, and zoib [35–37].

## Spatial descriptive analysis

A database with georeferenced aerial photographs provided by the Company for Urban Development of the State of Bahia (CONDER) was constructed in QGIS version 2.18.20. Photographs of the study site, with a scale of 1:2,000 and spatial resolution of 16 cm, were taken in 2006. The study team identified households within the study site and marked their positions onto hard copy 1:1,500 scale maps. A survey was conducted during the period of April 2015 to August 2016 to geocodify the location of the households and traps.

Kernel Density Estimation analysis (KDE) was performed to evaluate smoothed spatial distributions of sites of the three rodent capture campaigns, considering as a weight factor in the analysis the success rate of capture [38]. We use a distance matrix analysis to measure the shortest distance of households to an asphalted road >5 meters in width. To determine the smoothed population-adjusted risk distribution we calculated the ratio of the KDE for households that need rodent control to all households evaluated ('inspected'). The same analysis was performed for closed households relative to all households, as well as for households with the application of rodenticide relative to households that need rodent control.

## Results

### Chemical intervention

During the chemical intervention, all households in valleys 1 and 3 (N = 939) were visited, 283 in valley 1 and 656 in valley 3. Table 1 presents data on the coverage of the chemical intervention stratified by the valley. However, only two-thirds of the households (634, 67.5%) were inspected in order to evaluate rodent infestation. Closed households (32.5%) were the main reason why households were not inspected. Only 0.2% refused to participate.

Among the inspected households, 60% (380/634) showed signs of rodent infestation. Seventy-eight percent (297/380) of infested houses were treated with rodenticide. The remaining 22% (83/380) were not treated due to the presence of children or animals that could be at risk. Forty percent (n = 254) showed no signs of rodent infestation or environmental deficiencies and therefore did not qualify for intervention. However, 16% of these (n = 41) received rodenticide applications at the request of the resident.

Among households that received rodenticide applications, 67% received only paraffin blocks and 6% received only contact powder, while 27% received both types of rodenticides. The average number of paraffin blocks per household was 3.2, although there was a high variation in the intensity of application (range from 1 to 21). Only 12.3% of the households visited, which required application of rodenticide, received the three applications (Table 1) that are recommended for rodent control guidelines in urban settings.

There was high spatial heterogeneity in the distribution of households with infestation or environmental conditions that required the application of rodenticide. We also observed high heterogeneity in the application of rodenticides (Fig 2B).

During the second and third applications of rodenticide, it was not possible to estimate the bait consumption adequately, since the majority of them (73%, 500/685 in the first application) were not found in the following applications. Of the baits found during the second application (185), 96 (51.9%) were totally consumed, 37 (20%) partially consumed and 52 (28.1%) were not consumed.

Regarding rodent infestation as evaluated by CCZ, the pre-intervention survey revealed 39% of the households with signs of rodent infestation. In the post-intervention evaluation, this proportion dropped significantly to 21.1% (p <0.05), indicating a 54% reduction. The availability of resources such as water and food did not show a significant reduction in the environmental assessments carried out by CCZ. As examples, the proportion of households with available water decreased from 44.1% to 40%, while the proportion of households with available food increased from 37.2% to 39.7%.

### Rodent trapping

During the three trapping campaigns performed in valleys 1, 2, and 3 (one pre- and two post-intervention), 126 individuals of *Rattus norvegicus* and two of *Rattus rattus* were trapped in the 40 household points sampled in the three valleys (see the summary of the sampling in Table 2). During the pre-intervention campaign, trapping effort was 310 trap nights which

**Table 1. The proportion of inspections households per intervention site (valley 1 and 3).**

| Valley | Total households (n) | With inspection | House in need of treatment (total inspected) | Number of treatments | | | |
|---|---|---|---|---|---|---|---|
| | | | | 0 | 1 | 2 | 3 |
| 1 | 283 | 177 (62,5) | 130 (73,4) | 67 (37,8) | 50 (28,2) | 44 (24,8) | 16 (9) |
| 3 | 656 | 457 (69,7) | 341(74,6) | 248 (54,2) | 112 (24,5) | 70 (15,3) | 27 (5,9) |
| Total | 939 | 634 (67,5) | 471 (74,3) | 315 (49,7) | 162 (25,5) | 114(18) | 43 (6,8) |

**Table 2. Summary of population characteristics of *R. norvegicus* before and after chemical intervention in Pau da Lima, Salvador, Brazil.**

| | Pre-intervention | | Post-intervention | | | |
| --- | --- | --- | --- | --- | --- | --- |
| | Oct-Dec 2014 | | Nov–Dec 2015 | | Apr-May 2016 | |
| Valleys | 1 and 3 | 2 | 1 and 3 | 2 | 1 and 3 | 2 |
| No. of rats | 31 | 12 | 32 | 15 | 25 | 11 |
| Males | 20 (64.5) | 7 (58.3) | 16 (50) | 7 (46.7) | 10 (40) | 4 (36.4) |
| Females | 11 (35.4) | 5 (41.7) | 16 (50) | 8 (53.3) | 15 (60) | 7 (63.6) |
| Mass mean (SMI) | 264.9 | 248.9 | 212.2 | 211.9 | 237.8 | 221.5 |
| Males | 272.5 | 242.5 | 195.2 | 212.1 | 206.6 | 221.9 |
| Females | 251.2 | 257.8 | 229.2 | 211.6 | 258.6 | 221.3 |
| Age mean (days) | 82.77 | 108.9 | 91.67 | 81.2 | 84.25 | 99 |
| Males | 87.82 | 97.61 | 98.55 | 89.2 | 90.89 | 131 |
| Females | 69.03 | 124.8 | 84.8 | 74.1 | 79.84 | 80.6 |
| No. sexually active males | 17 (85) | 7 (100) | 14 (87.5) | 7 (100) | 8 (53.3) | 4 (100) |
| No. pregnant rats * | 3 (37.5) | 2 (100) | 9 (75) | 4 (80) | 4 (50) | 1 (25) |
| Lactating rats* | 6 (75) | 0 (0) | 8 (66.7) | 2 (25) | 5 (62.5) | 3 (75) |
| Pregnant lactating | 2 (33.3) | 0 (0) | 5 (62.5) | 1 (50) | 1 (20) | 0 (0) |
| Trap Success | 0.194 | 0.148 | 0.196 | 0.167 | 0.164 | 0.127 |

*Considering only sexually active females. The values in brackets are represented in percentages (%).

resulted in 43 *R. norvegicus* caught at 23 points. During the campaign performed three months after the intervention 47 *R. norvegicus* and 2 *R. rattus* were trapped at 21 points out of an effort of 312 trap nights. Finally, in the campaign that was carried out six months after the intervention, 36 *R. norvegicus* were trapped at 24 points, with a trapping effort of 324 trap nights. In this case, only *R. norvegicus* data was analyzed.

The rodent infestation index (PII) as evaluated at the 40 points did not show a reduction between the period before the intervention and the two periods after the chemical intervention (70%, 72%, and 65%, respectively). No significant differences were identified in mean trapping success between valleys or between pre-and post-intervention campaigns (Table 2).

As no pre and post-intervention differences were identified between valleys, and the chemical intervention was not homogeneous in the study area (Fig 3), trapping success comparisons were performed by point to evaluate if the number of baits had an influence on the trapping success at each point. The beta regression pointed to there being no relation between the number of baits applied and the capture success at each point. However, a positive relationship between capture success and three applications of contact powder was detected, but the number of houses that received the three applications of this rodenticide was very low (12.3%) and also had a greater number of signs of rodent activity (see the supplementary material).

The demographic characteristics of the pre-and post-intervention rodent populations are described in Table 2. Among males, 88.8% (56/63) were sexually active, in females at least one characteristic of sexual activity was observed in 61.9% of animals (39/63). Of the sexually active females, 58.9% (23/39) were pregnant and 61.5% (24/39) were lactating (37.5% of the lactating females were also pregnant). Among the pregnant females, the mean number of embryos was 11 (IQR 9–12).

No significant differences were found in the proportion between males and females, between the valleys (which received the chemical intervention or not) and between the pre and post-intervention campaigns (p> 0.05). There was no significant difference detected by the ANOVA with permutations in mean SMI between campaigns or valleys (Iter = 869, P = 0.485). In addition, no significant differences were detected in the number of embryos per pregnant female (ANOVA with permutation, Iter = 1.113, P = 0.354), the mean age

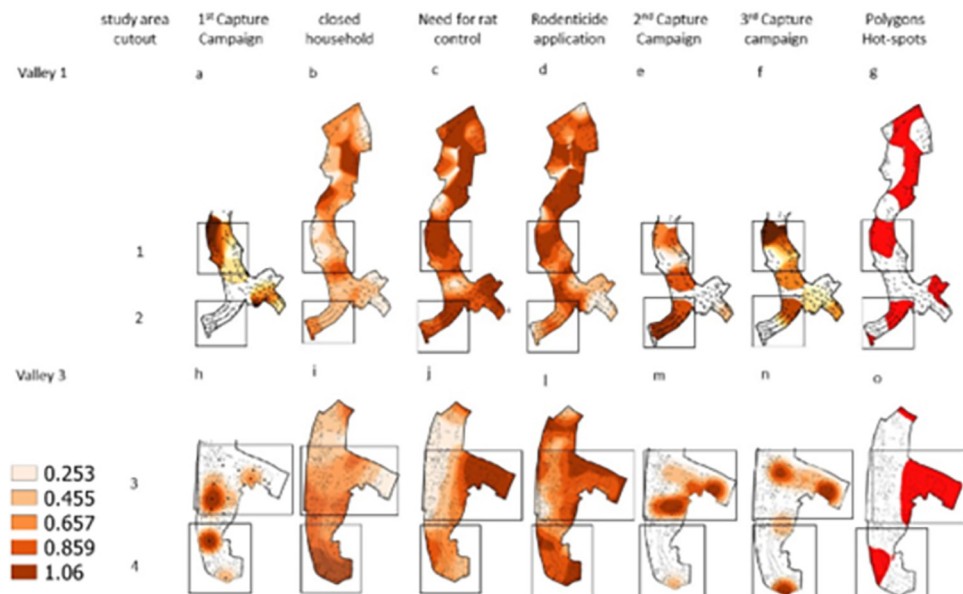

**Fig 3. Valley 1 and Valley 3 respectively: a and h) kernel Density Estimation (KDE) of the 1st rodent capture campaign, b and i) Kernel Ratio (KR) of closed households by the total of households, c and j) KR of households that need rodenticide control by evaluated households, d and l) KR of rodenticide application in households by households that need for rodent control, e and m) KDE of the 2nd rodent capture campaign, f and n) KDE of the 3rd rodent capture campaign, g and o) Polygons of hot areas of households that need for rodent control identified by agents of the Zoonoses Control Center of Salvador, Bahia, Brazil.** In valley 1 and valley 3 the upper part of the area was not shown in some figures due to the absence of rodent capture campaign at this site.

(Iter = 1.091, P = 0.365) or average age of pregnant females (valleys 1 and 3: Iter = 368, P = 0.527, 2: iter = 295, P = 0.152) between campaigns and valleys with and without chemical intervention. Finally, no significant result was found for the variation in the proportion of pregnant females between campaigns and valleys with or without intervention (p> 0.05).

## Spatial distribution rodenticide application analysis

Rodenticide intervention is designed to reduce abundance, with the success of rodent capture as a proxy. However, it was observed in the valleys 1, 2, and 3 that there was no influence of rodenticide application on the areas with a concentration of rodent captures. In sites 1 and 3, the areas of rodenticide application (Fig 3C and 3J) overlapped some of the concentration areas of the first campaign, which preceded the intervention. After the intervention, there was no observed reduction of areas with the concentration of capture of rodents. In site 2, a new area of capture concentration appeared in the two campaigns after the intervention, even with a record of application of rodenticide in this location (Fig 3C). Only in site three was it observed that after the intervention there was a reduction of rodent capture concentration. The closed households reached 37.8% in valley 1 (107/283) and 30.0% in valley 3 (197/656) and formed spatial aggregation in the area (Fig 3B and 3I), which may have influenced the intervention process.

## Discussion

Few previous studies have reported data on the efficiency of control programs in affecting urban populations of rodents [17,38,39]. Additionally, none of them systematically and longitudinally assessed the quality of the intervention in terms of coverage, intensity, and the

number of visits performed in each household in the studied area. In the present study, no changes were identified in the demographic characteristics of *R. norvegicus* populations before and after a chemical intervention carried out by the CCZ in Pau da Lima. Also, most of the spatial analyses did not present a negative influence of areas with the application of rodenticide on the areas with a successful capture. Likewise, we did not observe differences in the proportion of infested households during the application of rodenticide in a sample of the households evaluated. These findings contrast with the results obtained by the CCZ in the same intervention, which indicated a reduction in the proportion of households with signs of rodent infestation of 54% among all households evaluated during the intervention. The agents of the Zoonoses Control Center identified households that required intervention against rodents. This identification generated clusters and part of these clusters overlap areas with a concentration of rodents (Fig 3). The main challenges encountered by the CCZ during the implementation of the intervention were moderate household access (67.5%) and the low completion rate (12.3%) of the protocol, which includes three applications of rodenticide among the households in need of intervention. These results are key to identifying the barriers that are behind the low effectiveness in the control of synanthropic rodents and the prevention of zoonotic diseases, such as leptospirosis, in Brazil and other developing countries.

The relative abundance of rodents did not show significant differences between pre and post-intervention periods. We also did not find the correlation between the number of baits applied with the relative abundance (trap success) estimated at each evaluated point. These results are consistent with a previous study carried out in São Paulo [39], which used a similar methodology where there were no changes in the relative abundance of *R. rattus* after an intervention using rodenticides with the same active principles and formulations as those used in our study. In other intervention studies that combined measurements of sanitation, environmental education, and chemical control, a reduction in rodent activity/abundance was observed in poor communities in the cities of Buenos Aires (Argentina) and São Paulo (Brazil), and within ninety days the populations had not yet returned to pre-intervention levels [17,40]. In developed temperate countries where these methodologies were conducted, studies show a 50–90% decrease in the relative abundance of rodents, as reported in Baltimore [15]. This may be related to the availability of food throughout the year and high breeding rates in tropical environments where climate-related ecological limitans are relaxed [23].

In the evaluations carried out by the research group, no differences were observed in the proportion of households with signs of infestation. This contrasts with the CCZ results during the intervention, which reported a 54% drop in the proportion of infested households (from 39% to 21%). The divergence between the research group and the CCZ assessment can be attributed to differences at the time of evaluation, and the number of households evaluated.

In addition, we did not observe changes in the demographic characteristics of the population of rodents–sex ratio, age, SMI, and the number of embryos–after the chemical intervention. As far as we know, no previous study has described these population characteristics in detail in urban areas after chemical control to compare with our results. However, in Baltimore [41], when analyzing the composition of stationary, growing, and decreasing populations of *R. norvegicus*, it was observed that after a removal campaign, the rats of growing populations gained weight faster than those in stationary populations, even if younger. Thus, a population consisting mainly of young individuals in the post-intervention period would have been expected if control had reduced the abundance of the rodent population in Pau da Lima, which was not observed. On the other hand, no difference in sex ratio after removal was found, always remaining close to 1: 1 [41], and these results were similar to those observed in our study.

It is expected in small mammals that fecundity increases after a decrease in population density, given that fecundity and recruitment are regulated by density [42]. This phenomenon was observed in urban populations of *R. norvegicus* in Baltimore as an increase in pregnant females was recorded [43]. We did not observe differences in the proportion of pregnant females or the number of embryos, again suggesting that the intervention did not have an impact on population decrease. Alternatively, it would have been expected that after control, the average age of the pregnant females would decrease if the pregnancy rate remained constant [44], which was also not observed in our study.

The CCZ's chemical control campaign faced challenges common to other community-based strategies for controlling vectors or reservoirs [45]. The proportion of inspected households was moderate (67.6%). One of the main limitations of the program was the number of inaccessible houses (32.4%). The proportion of houses where intervention was not conducted is similar to that reported in rodent control programs elsewhere in Brazil, in São Paulo and Recife (Masi personal communication) and dengue control programs worldwide [46–48]. The visiting hours of the control agents were restricted to the morning shift, excluding the possibility that homes, where people were not available during this period, would receive visits from the agents and, consequently, the applications of rodenticides. The present study records 32.3% (304/939) of closed residences, which were not evaluated and formed areas of concentration (Fig 3b & 3i), which may influence intervention actions. Changes in visiting times/schedules may be an alternative to increase program efficiency [46]. However, the logistics and cost of these changes should be considered.

Another fact observed in this study was the heterogeneous distribution of rodenticide application in the study area (Fig 2B). Some areas received more coverage while others had little or no treatment. Additionally, only 12% of the households that had signs or environmental characteristics associated with rodent infestation had the complete intervention scheme (3 applications of rodenticide). Unfortunately, previous studies do not report data on the completeness of reported interventions [17,39].

There are several factors that might be playing into the lack of success of the interventions as rodent population control: baits might be poorly deployed, could be eliminated from the environment (either by being picked up by an animal or human, or displaced by environmental factors) or the baits chosen are being ignored due to the presence of more palatable foodstuffs [49], or due to neophobic/bait resistance behaviors [50,51]. It is also possible that local rodent populations are developing resistance against rodenticides [52]. It is possible that baits might lose palatability with time and exposure to the elements [53].

This study has important limitations. The measurements used to evaluate the effectiveness of the program by the research team and CCZ are necessarily comparable, however, since it is known that there is no relevant correlation between relative abundance estimated by capture methods and infestation levels evaluated by signs [54]. In addition, the pre-intervention evaluation carried out by the research group was conducted seven months before the chemical intervention campaign. Changes in environment or rodent population could have occurred between the pre-intervention evaluation and intervention. However, there was a prior expectation that the rodent population is temporally stable in the study area, in the absence of interventions [23].

The evidence presented above suggests that the current intervention design based mainly on rodenticide application was not efficient in reducing the abundance of rodents in this community. It is known that the methodology proposed by the Ministry of Health to control rodents includes, in addition to the application of rodenticides, measures of sanitation and environmental management as well as educational actions [12,13]. However, due to the operational restrictions faced by most of the rodent control programs in Brazil and other countries,

this control is done using mainly chemical rodenticides, which has not been an effective long-term solution [5]. The use of rodenticides often is used as a short-term measure against the lack of structure to fight against neglected diseases, such as leptospirosis [21]. In addition, inefficient use of rodenticides may generate a condition for rapid reproduction by surviving rodents and the perpetuation of genetic qualities associated with resistance [55]. In addition, educational actions, without environmental improvements, also do not seem to have an effect on residents' practices. In this study, even with the recommendations of endemic agents during the intervention, the availability of water and food for rodents remained the same before and after educational actions.

The control of rodents in urban areas of Brazil faces a series of restrictions, mainly operational, such as a lack of human and financial resources that do not allow the maintenance of routine actions. These programs are negatively affected by outbreaks of other more visible diseases such as dengue and Zika. Other challenges are the political instability that leads to rotation in leadership positions in the control bodies, the difficulty of access to some places with high levels of violence and drug trafficking, the lack of access and cleaning of vacant lots and houses, and the lack of a continuous training plan for CCZ. These difficulties can be overcome in part by the availability of technical support related to control methods and local ecological factors, which is accessible to endemic agents working in the field, and by carrying out integrated activities with the implementation of sanitation infrastructure.

The importance of a programmatic approach to urban rat population management that incorporates long-term planning, programming, data management, and mapping capabilities is clearly needed to address the task of rodent control [56–58]. The CCZ of Salvador, for example, does not have a team using Geographic Information System (GIS) tools, important in the planning and management of actions for this type of entity. For these purposes, municipal management of zoonoses control must be closely associated with pest management science, and efforts must be made at the political and administrative levels to ensure that zoonoses control programs are associated with universities and research institutions and can be better designed, implemented, and sustained. Some of the few examples in the literature of successful control programs were coordinated in Budapest, Hungary [59,60], and Kuwait [61], and all emphasized the importance of investments in sanitation and environmental management in conjunction with the rodent control measures mentioned above.

Evaluation of the effectiveness of control programs on the population of rodent reservoirs of zoonoses is essential for the prevention of human diseases affecting the most vulnerable communities in developing countries. These evaluations allow the identification of barriers in rodent control programs as in the coverage pattern and intensity of the interventions, allowing direct actions to overcome these limitations. Interventions to improve environmental deficiencies, for example, closing open sewers, and implementing garbage collection, should also be integrated into chemical control, since in these areas only chemical control seems insufficient to reduce the abundance of rodents and consequently, the transmission of rodent-borne zoonosis.

## Supporting information

**S1 File.**
(DOCX)

## Acknowledgments

We would like to thank the staff of Fiocruz and Zoonoses Control Center of Salvador for their help in carrying out this study. Mayara Carvalho, Leonardo Ferreira and Nivison Júnior for

their support in database management and Priscilla Machado and Djime Dourado for the construction of the maps, Hussein Khalil for his support in statistical tests and Renato Reis for his support in spatial analysis. We would also like to thank the associations of residents, community leaders and residents who constitute the Urban Health Council of Pau da Lima.

## Author Contributions

**Conceptualization:** Ricardo Lustosa, Hussein Khalil, Mitermayer G. Reis, James Childs, Mike Begon, Federico Costa.

**Data curation:** Arsinoê Cristina Pertile, Mitermayer G. Reis, Albert I. Ko, Federico Costa.

**Formal analysis:** Arsinoê Cristina Pertile, Ticiana Carvalho-Pereira, Caio Graco Zeppelini, Fábio Neves Souza, Daiana S. Oliveira, Hussein Khalil, Mitermayer G. Reis, James Childs, Mike Begon, Federico Costa.

**Funding acquisition:** Federico Costa.

**Investigation:** Arsinoê Cristina Pertile, Ticiana Carvalho-Pereira, Gabriel Ghizzi Pedra, Jesus Alonso Panti-May, Udimila Oliveira.

**Methodology:** Arsinoê Cristina Pertile, Ricardo Lustosa, Ticiana Carvalho-Pereira, Gabriel Ghizzi Pedra, Jesus Alonso Panti-May, Udimila Oliveira, Caio Graco Zeppelini, Daiana S. Oliveira, Mitermayer G. Reis, James Childs, Albert I. Ko, Mike Begon, Federico Costa.

**Project administration:** Federico Costa.

**Resources:** Udimila Oliveira, Albert I. Ko, Mike Begon.

**Software:** Ricardo Lustosa.

**Supervision:** Albert I. Ko, Federico Costa.

**Validation:** Ricardo Lustosa, Udimila Oliveira, Hussein Khalil, Albert I. Ko, Mike Begon, Federico Costa.

**Visualization:** Ricardo Lustosa, Caio Graco Zeppelini, Fábio Neves Souza, Daiana S. Oliveira.

**Writing – original draft:** Arsinoê Cristina Pertile, Ricardo Lustosa, Ticiana Carvalho-Pereira, Gabriel Ghizzi Pedra, Jesus Alonso Panti-May, Fábio Neves Souza, Daiana S. Oliveira, Hussein Khalil, Mitermayer G. Reis, James Childs, Federico Costa.

**Writing – review & editing:** Caio Graco Zeppelini, James Childs, Albert I. Ko, Mike Begon, Federico Costa.

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
