## [Decision Letter · Decision Letter 0]

8 Mar 2022

PONE-D-21-36428Evaluation of the impact of chemical control on the ecology of Rattus norvegicus of an urban community in Salvador, BrazilPLOS ONE

Dear Dr. Zeppelini,

Thank you for submitting your manuscript to PLOS ONE. After careful consideration, we feel that it has merit but does not fully meet PLOS ONE’s publication criteria as it currently stands. Therefore, we invite you to submit a revised version of the manuscript that addresses the points raised during the review process.

We look forward to receiving your revised manuscript.

Kind regards,

Bi-Song Yue, Ph.D

Academic Editor

PLOS ONE

Journal Requirements:

"no competing interests"

 This information should be included in your cover letter; we will change the online submission form on your behalf

Reviewers' comments:

Reviewer's Responses to Questions

**Comments to the Author**

1. Is the manuscript technically sound, and do the data support the conclusions?

Reviewer #1: Partly

Reviewer #2: Yes

Reviewer #3: Yes

2. Has the statistical analysis been performed appropriately and rigorously? 

Reviewer #1: Yes

Reviewer #2: Yes

Reviewer #3: Yes

3. Have the authors made all data underlying the findings in their manuscript fully available?

Reviewer #1: Yes

Reviewer #2: Yes

Reviewer #3: Yes

4. Is the manuscript presented in an intelligible fashion and written in standard English?

Reviewer #1: Yes

Reviewer #2: Yes

Reviewer #3: Yes

5. Review Comments to the Author

Reviewer #1: The authors evaluated the success of a chemical intervention of the CCZ (Center for Control of Zoonosis, Salvador de Bahia, Brazil) to control rat infestation in a neighbourhood of 3,700 people. The intervention was justified according to potential transmission of Leptospira interrogans from rats to humans, which causes severe disease (10,000 cases in Brazil, according to authors). Giving the Brazil human population (more than 200 million) this means a prevalence of 0.05% of severe disease, and only two persons in the neighbourhood would be prone to severe disease according to that small rate. Readers do not know anything about actual prevalence of Leptospira in rats in the country, but this reduced rate of infection maybe is “a needle in a haystack”. Maybe no one in the neighbourhood studied suffers from severe leptospirosis.

The authors showed that the chemical intervention had no significant effects on rats’ populations, which can describe a possible case of resistance to rodenticides, but also some uncontrolled biases in the study. One of the main concerns of this study is the way of handling baits, since the setting scheme seems to be very exposed to other non-target animals, including humans and their pets. Baits should be placed inside boxes specially designed for rats, without the possibility of other species having access. This does not seem to be the case, and it was worrying the loss of 73% of baits without knowing their fate.

I think that there is important missing information to understand the biological cycles of both the rats and the bacteria, and their relationships with the humans. One of the main shortcomings of the study is that interventions to reduce rat infestation are not well justified. I mean, there is no information about the prevalence of Leptospirosis in the population of rats studied, which can be the trigger to perform an intervention by the Administration (CCZ). If you are fighting against a zoonosis, but you don’t know anything about the actual rates of prevalence in reservoirs, this has not very much sense. If the prevalence was low, maybe an intervention is not justified since it will be completely inoperative. If the population of reservoirs is high, but the prevalence of disease is low, the risk of infection to humans maybe low. I think that an assessment of the number of rats carrying Leptospira is necessary to know if an intervention can be justified, even solely based on the economic costs of the intervention. If you are collecting animals, you can perform tests for Leptospirosis in the rats. At least, citation of other studies in Brazil is necessary.

I would like to know why the intervention was in such a small study area (0.17 km2). There were notified many cases of Leptospirosis?. Is 3.5% of cases a Low-Medium-High prevalence of this disease? There were high previous rates of rat infestation?

Abstract: correct “…in three valleys (two )…”

Line 124: What are the 11 areas related to the three valleys finally studied? What means high risk of disease?

Line 137: use a different quotation of authors…”This intervention follows standard methodologies described by Davis, Casta (25)…”, use the form “This intervention follows standard methodologies (25)…”, change throughout the text. This form of citation is repeated sometimes and is wrong, because the number format avoids using the authors’ names, and you are using the names plus the numbers, which represents a double citation.

Line 142: Describe “abandoned”

Line 144: If questionnaires are used for the intervention…, how did you rate each questionnaire to classify them into intervention/non-intervention?

Line 154: You use two different baits? Powder and Klerat paraffin? How were the baits placed to be inaccessible to other animals? This can be a shortcoming since they were not used when potential risk to children and pets. I think that poison always needs to be placed inaccessible to humans and pets, i.e., within especial boxes only accessed by rats. If intervention is not performed because of risk, the success to reduce infestation maybe limited.

Line 175: Describe better the sampling protocol... Why 10 months before intervention? What is the normal breeding cycle in rats?

The protocol is very confusing, since it is formed by two different (independent) interventions. One by the CCZ which is based on detection of signs of rats, and the other performed by the authors based on trapping.

The trapping design is insufficiently described. Did you trap in the peak/low of the breeding cycle? Only two traps per site, but for how many days? Did you analyse probability of capture to deal with false negatives? At least in my country, rats are very intelligent and difficult to trap without some days of pre-baiting, with traps set during some days open with access to bait.

Line 191: what means a 10m-radius buffer? If only two traps were set, saying that they were placed 10m apart is enough, isn’t it?

Line 265: What are the reasons to use one, two, both kind of rodenticides in every household? Can this be a source of biases?

Line 274: What means that you observed high heterogeneity in the application of rodenticides? They were not applied properly?

Line 277: This is strange, you did not find the first bait in 73% of sites. This means that baits were placed without much care, and surely were found by non-target species. This can represent some kind of non-ethical praxis. Baits need to be placed in areas inaccessible to other animals and need to be found in next campaign and retired at the end of the intervention for human safety.

Line 309: “A positive relationship between capture success and three applications of contact powder was detected.” This means that when more poison used, more captures success? This has nonsense.

Figure 1: Can you include the surrounding of the valleys to have information of the geography of the whole area? May be in shaded colours?

Line 718: Table 2: Not completely seen

Reviewer #2: General Comments

The current paper is an important contribution to our understanding of the challenges associated with managing rodent populations amongst the urban poor in developing countries. The key finding is that rodenticide applications did not significantly reduce rodent populations, nor affect the demographic “machinery” of the target pest population. These are extremely important findings given the high risk of rodent borne infections for humans in urban poor villages.

At the end of the “Introduction” it would be helpful to state a hypothesis based on what you expected to find. Then this hypothesis/hypotheses should be the lead to the discussion.

The text needs to be tightened considerably; I have highlighted some examples of where text can be deleted. I am surprised by the number of grammatical errors given three of the co-authors are professors at Yale University and University of Liverpool. A revised submission needs to be carefully reviewed for grammatical accuracy by one of these co-authors.

The paper can be improved in its readership reach if there is discussion of research on rodent management in poor villages in Africa (see Taylor et al. (reference given in detailed comments)) and more recent European studies (see Walther et al 2021 https://doi.org/10.1016/j.scitotenv.2021.147520 and references therein). There also has been lot of research publications on ecologically-based rodent management since 1999. Given the ecological and demographic focus of this paper, then it may be useful to place the findings and future directions for research in this context.

Baseline rodent trapping: I am surprised that the baseline trapping was conducted in October 2014 and the first round of rodenticides was applied in July 2015. Why is there such a long gap between baseline data and application of the treatment? This is a major limitation to the study. The delay is not mentioned until line 442 during the discussions of shortcomings of the study. Even then there is no reason given for such a long delay. I am not convinced by the glib argument that “However, there was a prior expectation that the rodent population is temporarily stable in the study area, in the absence of interventions.” Moreover, it is not description of what rodent control measures, if any, were applied by residents prior to the baseline trapping.

Although the trapping protocols are covered in a previous paper, as a minimum there needs to be a description of the type and number of traps set each night, and their location. You cannot expect readers to be familiar with the previous paper published 6 years ago.

Results:

In the methods there is mention that the ‘Rodent body condition was estimated

using a "scaled mass index" (SMI)’. I did not see mention of SMI in the results.

Detailed feedback:

Line 94: ‘to less of 10%’ should be ‘to less than 10%’; ‘those population sizes’ change to ‘those populations’

Line 96: ‘a high a general’ delete the second ‘a’

Line 99: ‘and others,’ and other what? Demographic parameters?

Line 100: ‘conscripted’? please find a more suitable word.

Line 103: first use of R. norvegicus in the text therefore should be Rattus norvegicus

Line 155: entrance should be plural.

Line 160: What does ‘topic 2’ mean?

Line 193: age structur (spelling)

Line 249: evaluated (spelling)

Line 270: There is no Table 2. This information appears in Table 1!

Line 272: ‘It was possible to observe’, replace with ‘There was’

Line 274: Delete ‘as observed in valleys 1 and 3’.

Line 278: ‘Among the baits placed in the first application that were found’ change to ‘Of the baits found…’

Line 281: ‘Regarding to rodent infestation as evaluated by CCZ, the…’ change to The CCZ….

Line 298: ‘data was analized’ should be ‘data were analyzed’

Line 299: Delete ‘in the 40 points’

Line 303: There is no Table 3. This should be Table 2.

Line 335; ‘was not observed reduction of areas’ change to ‘was no observed reduction of area’

Line 336: Delete ‘in the following campaigns’ this is redundant.

Be clearer with use of ‘sites’ here and elsewhere (refer to Figure 3 if need be). For example, ‘In the site section 2,’ should simply be ‘In site 2,’

Line 342: ‘in the intervetion process’ should be ‘the intervention process’ (delete in and correct the spelling of intervention)

Line 343: ‘identifyed households that requiring’ change to ‘identified households that required’

Line 344: spelling of identification

Line 375: “than those” change to “as those”

Line 377: abundance spelling

Line 380: conducted spelling

Line 380: ‘these methodologies were contucted’ change to ‘similar methodologies…’

Line 381-383: Might there also be differences in the efficacy of the rodenticides because of increased genetic resistance. The Baltimore study was done in 1948! This point should be raised here rather that brief mention later (line 435). Another factor is possible behavioral resistance to rodenticide baits (e.g. Brunton CFA, Macdonald DW, Buckle AP (1993) Behavioural resistance toward poison baits in brown rats, Rattus norvegicus. Appl Anim Behav Sci 38:159–174.)

At some stage in the Discussion I expected a comparison of more relevant studies such as that by Taylor PJ, Arntzen L, Hayter M, Iles M, Frean J, Belmain SR. Understanding and managing sanitary risks due to rodent zoonoses in an African city: beyond the Boston Model. Integrative Zoology. 2008; 3: 38-50. https://doi.org/10.1111/j.1749-4877.2008.00072.x

Figures and Tables

- As indicated above there is no Table 2 and Table 3 is obviously Table 2.

- Table 1 requires more explanation in the title.

- The legend for Figure 3 appeared after Table 1, which was slightly confusing initially

Table 2 (not 3). I assume that the numbers in brackets are percentages.

‘Pos intervention’ change to ‘Post Intervention’

The last column of the Table is not complete in my version. For the April

sampling there are no data shown for Village 2.

Figure 3. Need to explain the introduction of ‘sites’ (1 to 4), which appear in the text.

References

There are references that are incomplete. Often there are no page numbers. Please be consistent with the use of issue numbers – some references have them, some do not. Usually, it is best not to include issue number. Please also be consistent with species names in italics.

Also, book chapters are variable in their style, and sometimes the names of the authors are incorrect. For example, the correct citation of Cavia et al 2012 is:

Cavia R, Cueto GR, Suárez OV (2012) Techniques to estimate abundance and monitoring rodent pests in urban environments. In: Integrated Pest Management and Pest Control—Current and Future Tactics, Larramendy ML, Soloneski S (editors). Rijeka: InTech, pp 147–172.

Given that PLOS ONE does minimal editing of accepted manuscripts, it is essential to pay particular attention to the required detail of references.

Reviewer #3: This manuscript aims to assess the effectiveness of rodenticide application campaigns in infested households in poor urban areas of San Salvador, Bahia.

The information come from two sources: the CCZ and the research team, which only performed infestation evaluation and captures for data about rats populations, as abundance and population structure, but did not designed the control application.

In general the manuscript is clear, the methods adequate and the results and discussion are consistent. Results are novel and relevant to public health concerning the transmission of leptospirosis and other zoonoses by rats.

I have some comments and doubts:

1- It is not clear when was conducted the pre infestation evaluation by CCZ, and why the capture evaluation was conducted 10 months before the intervention. Along 10 months rat populations may have changed their abundance, and in consequence the comparison according to the effect of chemical application is not valid. The authors, however, took this fact in account and discuss it.

2- Lines 216-220 are not clear.

3- It is not clearly stated how was defined “the need of rodenticide application”

4- line 344. It must be identification instead of indetification.

5- Is available information about the prevalence of rodenticide resistance in rats populations of the area?

6. PLOS authors have the option to publish the peer review history of their article (what does this mean?). If published, this will include your full peer review and any attached files.

Reviewer #1: No

Reviewer #2: No

Reviewer #3: No

---

## [Author Response · Author response to Decision Letter 0]

30 May 2022

We thank the reviewers for their thoughtful and constructive comments and suggestions. We are submitting our responses along with a revised manuscript, "Evaluation of the impact of chemical control on the ecology of Rattus norvegicus of an urban community in Salvador, Brazil", for consideration as a research article to be published in PLOS ONE.

Again, we appreciate your consideration in reviewing our manuscript and please contact me if there are any questions on the preparation of the revision.

Sincerely,

Caio Graco Zeppelini,

Corresponding Author

Reviewers' comments:

Reviewer's Responses to Questions

Comments to the Author

Reviewer #1: The authors evaluated the success of a chemical intervention of the CCZ (Center for Control of Zoonosis, Salvador de Bahia, Brazil) to control rat infestation in a neighbourhood of 3,700 people. The intervention was justified according to potential transmission of Leptospira interrogans from rats to humans, which causes severe disease (10,000 cases in Brazil, according to authors). Giving the Brazil human population (more than 200 million) this means a prevalence of 0.05% of severe disease, and only two persons in the neighbourhood would be prone to severe disease according to that small rate. Readers do not know anything about actual prevalence of Leptospira in rats in the country, but this reduced rate of infection maybe is “a needle in a haystack”. Maybe no one in the neighbourhood studied suffers from severe leptospirosis.

The authors showed that the chemical intervention had no significant effects on rats’ populations, which can describe a possible case of resistance to rodenticides, but also some uncontrolled biases in the study. One of the main concerns of this study is the way of handling baits, since the setting scheme seems to be very exposed to other non-target animals, including humans and their pets. Baits should be placed inside boxes specially designed for rats, without the possibility of other species having access. This does not seem to be the case, and it was worrying the loss of 73% of baits without knowing their fate.

I think that there is important missing information to understand the biological cycles of both the rats and the bacteria, and their relationships with the humans. One of the main shortcomings of the study is that interventions to reduce rat infestation are not well justified. I mean, there is no information about the prevalence of Leptospirosis in the population of rats studied, which can be the trigger to perform an intervention by the Administration (CCZ). If you are fighting against a zoonosis, but you don’t know anything about the actual rates of prevalence in reservoirs, this has not very much sense. If the prevalence was low, maybe an intervention is not justified since it will be completely inoperative. If the population of reservoirs is high, but the prevalence of disease is low, the risk of infection to humans maybe low. I think that an assessment of the number of rats carrying Leptospira is necessary to know if an intervention can be justified, even solely based on the economic costs of the intervention. If you are collecting animals, you can perform tests for Leptospirosis in the rats. At least, citation of other studies in Brazil is necessary.

I would like to know why the intervention was in such a small study area (0.17 km2).

There were notified many cases of Leptospirosis?. Is 3.5% of cases a Low-Medium-High prevalence of this disease? There were high previous rates of rat infestation?

Line 124: What are the 11 areas related to the three valleys finally studied? What means high risk of disease?

Response: the 11 areas represent priority points detected in the entire territory of Salvador, of which the community here studied is one of them. High risk areas were detected based on the historical records of incidence, as well as the occurrence of the main risk exposure phenomena (such as rat complaints, floods). The text has been amended to better convey the information.

Line 142: Describe “abandoned”

Response: abandoned in the literal dictionary sense, the houses are no longer inhabited or sought after as such.

Line 154: You use two different baits? Powder and Klerat paraffin? How were the baits placed to be inaccessible to other animals? This can be a shortcoming since they were not used when potential risk to children and pets. I think that poison always needs to be placed inaccessible to humans and pets, i.e., within especial boxes only accessed by rats. If intervention is not performed because of risk, the success to reduce infestation maybe limited.

Response: this was part of the internal protocol of the CCZ and was, at the time of the study, beyond the realm of our possibilities to propose any alterations. This is an aspect that has been part of the discussion between our group and the CCZ in order to design better interventions

Line 144: If questionnaires are used for the intervention…, how did you rate each questionnaire to classify them into intervention/non-intervention?

Response: areas of intervention were classified according to the performance of the CCZ, the chemical intervention conducted by the CCZ was carried out in two areas (valleys 1 and 3). Valley 2 was classified as control (non-intervention), as this area would not have CCZ action. Baseline infestation rates were carried out in all areas, but in the intervention areas this survey was carried out by the CCZ and in the control area it was carried out by our research team using questionnaires standardized for evaluating Building Infestation Index. The same has variables raised by the CCZ.

Line 175: Describe better the sampling protocol... Why 10 months before intervention? What is the normal breeding cycle in rats?

The protocol is very confusing, since it is formed by two different (independent) interventions. One by the CCZ which is based on detection of signs of rats, and the other performed by the authors based on trapping.

The trapping design is insufficiently described. Did you trap in the peak/low of the breeding cycle? Only two traps per site, but for how many days? Did you analyse probability of capture to deal with false negatives? At least in my country, rats are very intelligent and difficult to trap without some days of pre-baiting, with traps set during some days open with access to bait.

Response: We restructured the text according to the comments for a better understanding of the text.

Line 191: what means a 10m-radius buffer? If only two traps were set, saying that they were placed 10m apart is enough, isn’t it?

Response: the traps are places somewhere within a 10m buffer, with no indication of the distance between traps, as is in the text: “Tomahawk traps were installed in each household within a 10-meter radius buffer”,

Line 265: What are the reasons to use one, two, both kind of rodenticides in every household? Can this be a source of biases?

Response: These are decisions made internally by the CCZ team, beyond of our team’s scope of authority. The purpose of applying both contact powder and provide poison blocks is to try and maximize the chances of exposing the rats to a lethal dose of rodenticide (powder to try and reach those who might not reach or might display neophobia to the blocks, and the blocks for those residing outside of the area applied. While we recognize that, at a finer level of analysis, this could introduce a lot of noise to the assessment of the effects of rodenticides in particular, this limitation has been taken into account on our analytical design.

Line 274: What means that you observed high heterogeneity in the application of rodenticides? They were not applied properly?

Response: The high heterogeneity of rodenticide applications cited means that some areas received greater coverage while others had little or no rodenticide treatment. Only 12% of the households that had environmental signs or characteristics associated with rodents had the complete intervention schedule (which includes 3 applications of rodenticide). These differences are due to the difficulty of access by CCZ agents to some households, refusal of residents due to the presence of domestic animals in the house, or absence of a resident at home at the moment of the visit. In addition, sites within the entire study area remained inaccessible due to drug trafficking-related violence.

Line 277: This is strange, you did not find the first bait in 73% of sites. This means that baits were placed without much care, and surely were found by non-target species. This can represent some kind of non-ethical praxis. Baits need to be placed in areas inaccessible to other animals and need to be found in next campaign and retired at the end of the intervention for human safety

Response: we recognize this as a problem. However, observe the previous answers regarding the CCZ protocols. While there are provisions in their protocol to not apply rodenticide blocks in areas where there are children, pets or animal rearing (such as people raising hens) to minimize the risk of contact with non-targets, we do consider that a part of the baits might be removed as bycatch (likely by possums), or more likely being washed out by torrential rain.

Line 309: “A positive relationship between capture success and three applications of contact powder was detected.” This means that when more poison used, more captures success? This has nonsense.

Response: this has been observed in other studies we have conducted. Several possibilities for the phenomenon have been considered, and our current stance is that the relationship observed is not causal, but the areas being baited are the areas with higher rat presence/activity.

Figure 1: Can you include the surrounding of the valleys to have information of the geography of the whole area? May be in shaded colours?

Response: We agree with your suggestion, however, when including the surroundings of the shaded valleys to have information on the geography of the entire area, we did not obtain a quality view of the area, we decided to keep the figure with the areas of the valleys of studies highlighted in another figure.

Abstract: correct “…in three valleys (two )…”

Line 137: use a different quotation of authors…”This intervention follows standard methodologies described by Davis, Casta (25)…”, use the form “This intervention follows standard methodologies (25)…”, change throughout the text. This form of citation is repeated sometimes and is wrong, because the number format avoids using the authors’ names, and you are using the names plus the numbers, which represents a double citation.

Line 718: Table 2: Not completely seen

Response: corrections were made to the text as pointed.

Reviewer #2: General Comments

The current paper is an important contribution to our understanding of the challenges associated with managing rodent populations amongst the urban poor in developing countries. The key finding is that rodenticide applications did not significantly reduce rodent populations, nor affect the demographic “machinery” of the target pest population. These are extremely important findings given the high risk of rodent borne infections for humans in urban poor villages.

At the end of the “Introduction” it would be helpful to state a hypothesis based on what you expected to find. Then this hypothesis/hypotheses should be the lead to the discussion.

The text needs to be tightened considerably; I have highlighted some examples of where text can be deleted. I am surprised by the number of grammatical errors given three of the co-authors are professors at Yale University and University of Liverpool. A revised submission needs to be carefully reviewed for grammatical accuracy by one of these co-authors.

The paper can be improved in its readership reach if there is discussion of research on rodent management in poor villages in Africa (see Taylor et al. (reference given in detailed comments)) and more recent European studies (see Walther et al 2021 https://doi.org/10.1016/j.scitotenv.2021.147520 and references therein). There also has been lot of research publications on ecologically-based rodent management since 1999. Given the ecological and demographic focus of this paper, then it may be useful to place the findings and future directions for research in this context.

Baseline rodent trapping: I am surprised that the baseline trapping was conducted in October 2014 and the first round of rodenticides was applied in July 2015. Why is there such a long gap between baseline data and application of the treatment? This is a major limitation to the study. The delay is not mentioned until line 442 during the discussions of shortcomings of the study. Even then there is no reason given for such a long delay. I am not convinced by the glib argument that “However, there was a prior expectation that the rodent population is temporarily stable in the study area, in the absence of interventions.” Moreover, it is not description of what rodent control measures, if any, were applied by residents prior to the baseline trapping.

Although the trapping protocols are covered in a previous paper, as a minimum there needs to be a description of the type and number of traps set each night, and their location. You cannot expect readers to be familiar with the previous paper published 6 years ago.

Response: We restructured the text according to the comments for a better understanding of the text.

Results:

In the methods there is mention that the ‘Rodent body condition was estimated

using a "scaled mass index" (SMI)’. I did not see mention of SMI in the results.

Response: the SMI is used for the population characterization, see table 2.

Line 381-383: Might there also be differences in the efficacy of the rodenticides because of increased genetic resistance. The Baltimore study was done in 1948! This point should be raised here rather that brief mention later (line 435). Another factor is possible behavioral resistance to rodenticide baits (e.g. Brunton CFA, Macdonald DW, Buckle AP (1993) Behavioural resistance toward poison baits in brown rats, Rattus norvegicus. Appl Anim Behav Sci 38:159–174.)

Response: we have added a passage on behavioral resistance on the baits.

At some stage in the Discussion I expected a comparison of more relevant studies such as that by Taylor PJ, Arntzen L, Hayter M, Iles M, Frean J, Belmain SR. Understanding and managing sanitary risks due to rodent zoonoses in an African city: beyond the Boston Model. Integrative Zoology. 2008; 3: 38-50. https://doi.org/10.1111/j.1749-4877.2008.00072.x

Line 99: ‘and others,’ and other what? Demographic parameters?

Response: We restructured the text according to the comments for a better understanding of the text.

Detailed feedback:

Line 94: ‘to less of 10%’ should be ‘to less than 10%’; ‘those population sizes’ change to ‘those populations’

Line 96: ‘a high a general’ delete the second ‘a’

Line 100: ‘conscripted’? please find a more suitable word.

Line 103: first use of R. norvegicus in the text therefore should be Rattus norvegicus

Line 155: entrance should be plural.

Line 160: What does ‘topic 2’ mean?

Line 193: age structur (spelling)

Line 249: evaluated (spelling)

Line 270: There is no Table 2. This information appears in Table 1!

Line 272: ‘It was possible to observe’, replace with ‘There was’

Line 274: Delete ‘as observed in valleys 1 and 3’.

Line 278: ‘Among the baits placed in the first application that were found’ change to ‘Of the baits found…’

Line 281: ‘Regarding to rodent infestation as evaluated by CCZ, the…’ change to The CCZ….

Line 298: ‘data was analized’ should be ‘data were analyzed’

Line 299: Delete ‘in the 40 points’

Line 303: There is no Table 3. This should be Table 2.

Line 335; ‘was not observed reduction of areas’ change to ‘was no observed reduction of area’

Line 336: Delete ‘in the following campaigns’ this is redundant.

Be clearer with use of ‘sites’ here and elsewhere (refer to Figure 3 if need be). For example, ‘In the site section 2,’ should simply be ‘In site 2,’

Line 342: ‘in the intervetion process’ should be ‘the intervention process’ (delete in and correct the spelling of intervention)

Line 343: ‘identifyed households that requiring’ change to ‘identified households that required’

Line 344: spelling of identification

Line 375: “than those” change to “as those”

Line 377: abundance spelling

Line 380: conducted spelling

Line 380: ‘these methodologies were contucted’ change to ‘similar methodologies…’

Figures and Tables

- As indicated above there is no Table 2 and Table 3 is obviously Table 2.

- Table 1 requires more explanation in the title.

- The legend for Figure 3 appeared after Table 1, which was slightly confusing initially

Table 2 (not 3). I assume that the numbers in brackets are percentages.

‘Pos intervention’ change to ‘Post Intervention’

The last column of the Table is not complete in my version. For the April

sampling there are no data shown for Village 2.

Response: We restructured the text according to the comments for a better understanding of the text. We have modified the sentences following the reviewer’s suggestion. 

Figure 3. Need to explain the introduction of ‘sites’ (1 to 4), which appear in the text.

Response:We have modified following reviewer’s suggestion. 

References

There are references that are incomplete. Often there are no page numbers. Please be consistent with the use of issue numbers – some references have them, some do not. Usually, it is best not to include issue number. Please also be consistent with species names in italics.

Also, book chapters are variable in their style, and sometimes the names of the authors are incorrect. For example, the correct citation of Cavia et al 2012 is:

Cavia R, Cueto GR, Suárez OV (2012) Techniques to estimate abundance and monitoring rodent pests in urban environments. In: Integrated Pest Management and Pest Control—Current and Future Tactics, Larramendy ML, Soloneski S (editors). Rijeka: InTech, pp 147–172.

Given that PLOS ONE does minimal editing of accepted manuscripts, it is essential to pay particular attention to the required detail of references.

Response: the references have their information retrieved by DOI through EndNote, and contain the standard information deposited in the DOI registry, and presented under the guidelines of the Vancouver style. Information for items without DOI were extracted in its entirety from the original files. The references were re-checked on their data, and updated.

Reviewer #3: This manuscript aims to assess the effectiveness of rodenticide application campaigns in infested households in poor urban areas of San Salvador, Bahia.

The information come from two sources: the CCZ and the research team, which only performed infestation evaluation and captures for data about rats populations, as abundance and population structure, but did not designed the control application.

In general the manuscript is clear, the methods adequate and the results and discussion are consistent. Results are novel and relevant to public health concerning the transmission of leptospirosis and other zoonoses by rats.

I have some comments and doubts:

1- It is not clear when was conducted the pre infestation evaluation by CCZ, and why the capture evaluation was conducted 10 months before the intervention. Along 10 months rat populations may have changed their abundance, and in consequence the comparison according to the effect of chemical application is not valid. The authors, however, took this fact in account and discuss it.

2- Lines 216-220 are not clear.

3- It is not clearly stated how was defined “the need of rodenticide application”

4- line 344. It must be identification instead of indetification.

5- Is available information about the prevalence of rodenticide resistance in rats populations of the area?

Response: No information on the subject is available.

---

## [Editor Report · Decision Letter 1]

14 Jun 2022

Evaluation of the impact of chemical control on the ecology of Rattus norvegicus of an urban community in Salvador, Brazil

PONE-D-21-36428R1

Dear Dr. Zeppelini,

We’re pleased to inform you that your manuscript has been judged scientifically suitable for publication and will be formally accepted for publication once it meets all outstanding technical requirements.

Kind regards,

Bi-Song Yue, Ph.D

Academic Editor

PLOS ONE

---

## [Editor Report · Acceptance letter]

27 Jun 2022

PONE-D-21-36428R1 

Evaluation of the impact of chemical control on the ecology of *Rattus norvegicus* of an urban community in Salvador, Brazil 

Dear Dr. Zeppelini:

I'm pleased to inform you that your manuscript has been deemed suitable for publication in PLOS ONE. Congratulations! Your manuscript is now with our production department. 

Kind regards, 

on behalf of

Dr. Bi-Song Yue 

Academic Editor

PLOS ONE